# The Effects of Statins, Ezetimibe, PCSK9-Inhibitors, Inclisiran, and Icosapent Ethyl on Platelet Function

**DOI:** 10.3390/ijms241411739

**Published:** 2023-07-21

**Authors:** Assunta Di Costanzo, Ciro Indolfi, Sabato Sorrentino, Giovanni Esposito, Carmen Anna Maria Spaccarotella

**Affiliations:** 1Division of Cardiology, Cardiovascular Research Center, University Magna Graecia Catanzaro, 88100 Catanzaro, Italy; indolfi@unicz.it; 2Division of Cardiology, Department of Advanced Biomedical Sciences, University of Naples Federico II, 80134 Naples, Italy; giovanni.esposito2@unina.it (G.E.); carmenannamaria.spaccarotella@unina.it (C.A.M.S.)

**Keywords:** platelet, residual platelet reactivity, anticholesterolemic drugs, statin, ezetimibe, PCSK9i, inclisiran, icosapent ethyl, LDL-C, triglycerides, hypercholesterolemia

## Abstract

This review aims to examine the complex interaction between dyslipidemia, platelet function, and related drug treatments. In particular, the manuscript provides an overview of the effects of major hypolipidemic drugs on platelet function. Indeed, growing evidence supports the view that statins, ezetimibe, PCSK9 inhibitors, inclisiran, and icosapent ethyl also act as antithrombotics. It is known that platelets play a key role not only in the acute phase of coronary syndromes but also in the early phase of atherosclerotic plaque formation. The goal of cholesterol-lowering therapy is to reduce cardiovascular events. The direct effects of cholesterol-lowering drugs are widely described in the literature. Lowering LDL-c (low-density lipoprotein cholesterol) by 1 mmol/L results in a 22–23% reduction in cardiovascular risk. Numerous studies have examined the direct antithrombotic effects of these drugs on platelets, endothelium, monocytes, and smooth muscle cells, and thus, potentially independent of blood LDL-cholesterol reduction. We reviewed in vitro and in vivo studies evaluating the complex interaction between hypercholesterolemia, hypertriglyceridemia, platelet function, and related drug treatments. First, we discussed the role of statins in modulating platelet activation. Discontinuation of statin therapy was associated with increased cardiovascular events with increased ox-LDL, P-selectin, and platelet aggregation. The effect of PCSK9-I (inhibitors of proprotein convertase subtilisin/kexin type 9, PCSK9 involved in the degradation of LDL receptors in the liver) was associated with a statistically significant reduction in platelet reactivity, calculated in P2Y12 reaction units (PRU), in the first 14 days and no difference at 30 days compared to placebo. Finally, in patients with hypertriglyceridemia, the REDUCE-IT study showed that icosapent ethyl (an ethyl ester of eicosapentaenoic acid that reduces triglyceride synthesis and improves triglyceride clearance) resulted in a 25% reduction in ischemic events and cardiovascular death. However, to date, there is not yet clear clinical evidence that the direct antithrombotic effects of the drugs may have a beneficial impact on outcomes independently from the reduction in LDL-C or triglycerides.

## 1. Introduction

Atherosclerosis is the primary cause of cardiovascular deaths, regardless of gender [1]. It involves the gradual development of fatty streaks in arterial walls, which transform into atheroma and eventually into vulnerable plaques. These atherosclerotic plaques consist of a fibrous cap and a lipid core, primarily composed of oxidized low-density lipoprotein (Ox-LDL). The plaque’s stability relies mainly on the fibrous cap’s thickness (FCT), as the exposure of the core material to the blood can trigger platelet activation. A vulnerable plaque is defined as having an FCT of ≤65 μm. Plaque rupture and subsequent formation of an intraluminal thrombus are the most common causes of acute coronary syndromes [2].

It has been established that nonpharmacological and pharmacological interventions aimed at lowering cholesterol levels impact cardiovascular morbidity and mortality substantially. The history of hypercholesterolemic therapy starts in 1950 with the description of the cholesterol synthesis pathway. The evolution of hypolipidemic therapy has made it possible to achieve ambitious therapeutic targets in a short time (Figure 1). 

The direct effect of the drugs currently available is related to lowering LDL-C. Several studies have shown an average cardiovascular risk decrease of 22–23% per 1.0 mmol/L LDL-C reduction [3]. Statins inhibit HMG-CoA reductase, causing the reduction in endogenous cholesterol synthesis in the liver. Ezetimibe inhibits the absorption of dietary and biliary cholesterol by inhibiting the Niemann–Pick C1-like 1 (NPC1L1) protein expressed in intestinal cells and hepatocytes. PCSK9 is a hepatic protease that binds to the LDLR receptor on the surface of the liver and causes its degradation. PCSK9 inhibitors are monoclonal antibodies that reduce the expression of this protease, leading to increased LDL-C uptake. Inclisiran is a small interfering RNA (siRNA) that penetrates hepatocytes and blocks the translation of PCSK9 mRNA. Bempedoic acid acts similarly to statins by blocking cholesterol synthesis through the inhibition of adenosine triphosphate citrate lyase (ACLY). The rationale for the use of hypolipidemic drugs is mainly related to the stabilizing effect of atheromatous plaque, the decrease in the amount of the lipid core, and the increase in the thickness of the fibrous cap (Figure 2) [4].

As a result, the latest international lipid guidelines now strongly advocate for lipid-lowering therapy as a crucial component of primary and secondary prevention in patients with atherosclerotic cardiovascular disease (ASCVD). In addition to diet and maximum tolerated statin or ezetimibe therapy, several additional lipid-lowering medications are available, including two fully human monoclonal antibodies (mAbs) anti-proprotein convertase subtilisin/kexin type 9 (PCSK9), alirocumab and evolocumab, a small interfering RNA (siRNA) that prevents the hepatic synthesis of PCSK9, inclisiran, and a further novel nonstatin drug bempedoic acid, are recently approved as agents available for use in adult hypercholesterolemia who do not achieve target LDL-C [5]. Furthermore, the focus on anti-lipid therapy has been very high in recent years due to the prospect of medical therapy to reduce Lp(a) even for aortic stenosis [6]. 

In addition to lipid-lowering therapy, antithrombotic treatment also plays a critical role in the secondary prevention of complications in individuals with established ASCVD. Of interest, the general population’s response to antiplatelet therapy is variable. In fact, in some cases, there is a low platelet response to antiplatelets called residual platelet reactivity (RPR). Ideally, a platelet reactivity test could measure the patient’s response to antithrombotic treatment to have adequate platelet inhibition [7]. The randomized controlled trial GRAVITAS evaluated whether customized antiplatelet therapy by measuring platelet aggregation on whole blood reduces major adverse events in patients undergoing angioplasty with stent implantation. In the 2796 patients enrolled in the study, platelet reactivity was measured at 12–24 h, 30 days, and 6 months after stent implantation. Values ≥ 235 P2Y12 reaction units (PRU) identified patients with high reactivity. It was observed that achieving clopidogrel reactivity < 208 PRUs was associated with a lower risk of death from cardiovascular causes, nonfatal myocardial infarction (MI), or stent thrombosis [8].

## 2. Interaction between Atherosclerotic Plaque and Platelet Activity

In vivo and in vitro studies have shown that platelets adhere to the intact endothelium without proaggregating factors [9]. Endothelium-expressed P-selectin (CD62P) contacts the platelet receptor PSGL-1 (P-selectin glycoprotein ligand-1) and mediates reversible adhesion and rolling. Stable adhesion is mediated by integrins, a family of CAMs (cell adhesion molecules). Stable binding to the endothelium activates the formation of an inflammatory environment that predisposes the development of atherosclerotic lesions. Platelets firmly attached to the endothelium increase the release of CD40L on the platelet surface, which then releases soluble CD40L. Activated platelets also express P-selectin. CD40L binding on the surface of platelets and endothelium increases adhesion molecules such as E-selectin, VCAM-1, ICAM-1, and proinflammatory cytokines such as interleukin-8 (IL-8), interleukin (IL)-6, RANTES (regulated on activation normal T cells expressed and secreted), monocyte endothelial chemotactic protein-1 (MCP-1), and metalloproteinase (MMP). MMPs are involved in the remodeling and degradation of the extracellular matrix. In addition, platelets release IL-1β, which increases MCP-1 secretion. MCP-1 protein promotes the recruitment of monocytes, which adhere to the endothelium through the platelet chemotactic factor RANTES and binding P-selectins and, subsequently the integrins VCAM (vascular cell adhesion molecule) [10]. Activated platelets on the endothelial surface release alpha-granules and secrete PF4 (platelet factor 4), among other proteins. PF4 is involved in the differentiation of monocytes into macrophages. Furthermore, it was observed that PF4 and Ox-LDL were colocalized at atherosclerotic lesions. PF4 prevents LDL from binding to the macrophage receptor LDL-R. In this way, LDL is not degraded but remains trapped in the vascular space and is oxidized [11,12]. Ox-LDL then binds class A scavenger receptors (SR-A, typical of activated macrophages and smooth muscle cells), class B scavenger receptors (CD36, expressed on macrophages and cells in the liver, brain, and heart), and a third class typically expressed by lysosomes (CD68, expressed on cells associated with the immune system and bone marrow such as monocytes, macrophages, dendritic cells, and osteoclasts). These binding favors endocytosis and the formation of foam cells that promote plaque formation and growth [13]. In turn, Ox-LDL increases the local production of chemokines that attract monocytes and the production of oxidized LDL lectin-like receptor-1 (LOX-1). LOX-1 is expressed on endothelial cells and allows Ox-LDL accumulation (Figure 3) [14]. 

Several factors play a role in modulating the platelet response. It has been shown that inflammatory diseases can lead to alterations in platelet function [15]. Indeed, platelets play a major role in the pathogenesis of acute coronary syndromes. The most frequent cause of acute coronary syndrome (ACS) is plaque rupture (60%). The second mechanism of ACS is the erosion of fibrous plaques. In both cases, exposure of the lipid core to the blood is the initial mechanism of platelet aggregation and intracoronary thrombus formation (Figure 4).

Exposure to collagen and von Willebrand factor (vWF) results in platelet activation. Platelets adhere to the endothelium-expressed P-selectin via the platelet glycoprotein (GP) GPIbα and PSGL-1. Subsequently, stable adhesion is mediated by integrins to collagen. The attached platelets release ADP and thromboxane A2. The injured endothelium exposes tissue factors that trigger the coagulation cascade and thrombin activation. Platelet aggregation and thrombus formation are the results of the activation of three signal pathways on the surface of platelets: thrombin protease-activated receptor-1, TxA2-thromboxane receptor, and ADP-P2Y12 receptor pathways. In this way, numerous platelets are drawn to the site of injury, promoting thrombus formation [16,17,18].

Several studies have described the indirect effects of LDL-C reduction in atherosclerotic plaque and platelet aggregation. The direct effect of antilipemic therapies on platelet function has also been investigated. In vitro, incubation of platelets and lipoproteins has shown how high levels of LDL and VLDL enhance platelet activity. In particular, apoB-100, expressed by LDL and VLDL, binds and activates the LDL receptor expressed by platelets and alters signal transduction. Platelets become more sensitive to activating stimuli and acquire a hyperaggregability phenotype [19].

In addition, platelets with higher mean platelet volume (MPV) correlate with a higher risk of ischemic events and thrombosis [20]. Several studies have demonstrated a statistically significant association between a hyperlipidemic environment and platelet volume indices (PVI), including the mean platelet volume (MPV), platelet distribution width (PDW), and platelet large cell ratio (P-LCR) [21,22].

## 3. The Effect of Statins on Platelet Function

The protective effect of statins on cardiovascular events is well-known. Statins, or HMG-CoA reductase inhibitors, have as their main mechanism of action the reduction in endogenous cholesterol synthesis in the liver. The mechanism underlying the beneficial effect is mainly related to reducing the lipid core and increasing the fibrous cap (Figure 2). However, several studies have shown that the protective effect of statins does not depend entirely on LDL-C reduction but is associated with a pleiotropic action. At high doses, statins have been shown in experimental studies performed in our laboratory to reduce neointimal proliferation after vascular injury and left ventricular hypertrophy (LVH) in a pressure overload model [23,24,25]. Restenosis is one of the main mechanisms associated with percutaneous coronary intervention (PCI) failure of bare metal stents (BMS). Coronary stent placement triggers an inflammatory reaction and the differentiation of vascular smooth muscle cells (VSMCs), which migrate and form neointima. Intimal hyperplasia is associated with activation of the ras pathway. Ras protein is responsible for RAF and MAPKK activation (mitogen-activated protein kinase). This signal transduction pathway plays a central role in the activation and growth of VSMCs [23]. Statins have been shown to inhibit the ras-RAF-MAPKK pathway. In this way, statins inhibit VSMC proliferation in vitro and neointima formation in a rat model of vascular injury [24]. The ras protein ras-RAF1-ERK1/2 signaling pathway is involved in the mechanism of LVH. 

In addition, statins are involved in modulating platelet activation. A 2003 study showed that discontinuation of statin therapy was associated with an increase in cardiovascular events with an increase in Ox-LDL, P-selectin, and platelet aggregation observed 14 days after discontinuation [26]. Niessen et al. investigated the role of a transporter on the platelet surface, explaining the mechanism of action of statins. Hepatic uptake of platelets was associated with the transporter OATP1B1 (organic anion-transporting peptide 1B1). At the platelet level on the plasma membrane, the OATP2B transporter is expressed, a high-affinity uptake transporter for atorvastatin and rosuvastatin. Within the platelet, the mechanism of action of statins occurs through the class inhibition of HMGCR (3-hydroxy-3-methylglutaryl coenzyme A reductase). The result is a reduction in mevalonate formation. The study showed that absorption of atorvastatin reduced thrombin-induced calcium release, resulting in reduced aggregation. This effect was reversed with the addition of mevalonate [27].

Another mechanism underlying the antiplatelet effect of statins is iteration with different signal transduction pathways. Statins, in particular simvastatin, atorvastatin, and rosuvastatin, play an important role in the inhibition to activation of several pathways (Table 1). 

Statins are involved in the inhibition of P-selectin exposure after cell activation. This prevents the stable adhesion of platelets to fibrinogen, neutrophils, monocytes, and endothelium. They also inhibit the coagulation cascade through the inhibition of the protease-activated receptor-1 (PAR-1) and tissue factor (TF) pathways. PAR-1 is the thrombin receptor. Thrombin/PAR-1 binding on the platelet surface results in irreversible platelet aggregation and stimulates the production of proinflammatory and adhesion molecules. The PARIS study showed reduced receptors in patients treated with statins [28]. TF is responsible for the extrinsic coagulation pathway and is responsible for thrombus formation. Elevated levels of circulating microparticles (cMP) and TF-bearing cMP have been observed in patients with acute coronary syndrome (ACS) [29]. cMP are microvesicles released by vascular (platelets) and activated inflammatory (monocytes) cells. The amount of cMP reflects the severity of the infarction and the thrombotic load. A study by Suades et al. in patients with hypercholesterolemia treated with statins assessed the reduction in cMP [30].

A key mechanism of platelet-endothelial binding under inflammatory conditions is the binding between CD40 and CD40L. It is known that a hypercholesterolemic phenotype has an increased expression of CD40L. Several in vivo studies have shown a reduction in serum CD40L levels in statin-treated patients [31,32]. Statins activate the peroxisome proliferator-activated receptor (PPAR α and PPAR γ) pathway [33]. Activation of these receptors results in the inhibition of platelet function through the inhibition of protein kinase Cα (PKCα). Inhibited PKCα blocks the MAPK, PI3K-Akt-cGMP, Erk, and p38 signal transduction pathways that play an important role in platelet degranulation and aggregation [34]. Conversely, receptor activation increases cytoplasmic cAMP levels. The cAMP/PKA and cGMP/NO pathways are the major platelet inhibitory pathways, and there are numerous downstream targets responsible for the inhibitory effect on platelets. The cAMP activates protein kinase A (PKA) and phosphorylates the inositol triphosphate receptor (IP3R). Under normal conditions, aggregation is promoted by the release of calcium from the dense tubular system and the entry of Ca^2+^ from the extracellular space [35]. Phosphorylation of IP3R prevents calcium release from the dense tubular system with reduced intracytoplasmic Ca^2+^ availability [36]. Another mechanism of action of statins is the NO-dependent protective effect. Statins are implicated in the NO production pathway through the inhibition of geranylgeranyl pyrophosphate (GGPP), an intermediate of the cholesterol pathway. GGPP is involved in the inactivation of the rho protein. Activated rho protein causes instability of eNOS mRNA, blocking its function. According to in vitro and in vivo tests, statins instead cause an increase in eNOS expression and NO production in endothelial cells. NO is associated with a vasoprotective, vasodilating effect, and reduced platelet aggregation [37]. It has also been shown to reduce oxidative stress. The antioxidant effect of statins is associated with the downregulation of the NOX2 enzyme (subunit of a complex that produces ROS) and NADPH involved in the production of isoprostanes [38]. Another mechanism underlying the reduction in oxidative stress is the downregulation of phospholipase A2 (PLA2). Under normal conditions, reactive oxidant species activate PLA2 and cause the release of arachidonic acid (AA). Subsequently, AA is converted by COX1 into prostaglandin H2 (PGH2), which is then converted by appropriate synthases into thromboxane A2 (TxA2) [39]. Statins inhibiting the PLA2 pathway cause a reduction in TxA2 [40]. This mechanism could hypothesize a synergistic action in the coadministration of statins and aspirin [41].

## 4. The Effect of Ezetimibe on Platelet Function

The mechanism of action of statins results in the inhibition of HMG-CoA reductase. This inhibition results in increased expression of hepatic LDL-C receptors and reduction in LDL-C. Intestinal feedback is therefore activated, resulting in increased intestinal absorption of dietary cholesterol. Ezetimibe inhibits the uptake of dietary and biliary cholesterol. The drug’s target is the Niemann–Pick C1-like 1 (NPC1L1) protein expressed in intestinal cells and hepatocytes. At the intestinal level, it reduces the absorption of exogenous cholesterol. At the hepatic level, the reduced absorption of cholesterol increases the expression of LDL receptors with increased hepatic uptake of LDL-C. The effect is a reduction in circulating cholesterol by approximately 10–15%. The addition of ezetimibe in therapy has the advantage of reaching lower LDL-C targets faster and also allows lower statin dosages to be used.

The results of the IMPROVE-IT study demonstrated the efficacy of statin/ezetimibe combination therapy compared to monotherapy. A reduction in LDL-C levels and a further 9% reduction in cardiovascular events (cardiovascular death, acute myocardial infarction, and stroke) were observed one year after randomization [42]. In vitro, studies have demonstrated a direct effect of ezetimibe on endothelium and platelets [43,44]. Ezetimibe directly attenuates platelet activation and has significant endothelial cell-mediated effects on selected markers of atherosclerosis. In particular, ezetimibe has been shown to reduce the expression of urokinase-type plasminogen activator receptor (uPAR or CD87) on endothelial cells [45]. The uPAR plays a key role in the mechanism of platelet adhesion and aggregation: uPAR/uPA binding (expressed by platelets) results in fibrin production and thrombus formation. Furthermore, platelets incubated with ezetimibe showed reduced expression of CD40L and the selectin CD62P [44]. Hussein et al. demonstrate that ezetimibe reduced platelet aggregation by reducing platelet peroxidation. Ox-LDL improve platelet aggregability. By reducing LDL levels, ezetimibe resulted in an indirect reduction in platelet activation [44]. 

Furthermore, a study by Tano et al. described in 42 patients the protective effect of ezetimibe as monotherapy in patients with dyslipidemia by assaying cholesterol and platelet-activating factor acetylhydrolase (PAF-AH) activity. PAF-AH is a circulating biomarker indicative of inflammation and a predictor of coronary artery disease. Ezetimibe shows a reduction in PAF-AH [45]. Hussein et al. demonstrated a reduction in platelet-activating factor acetylhydrolase (PAF-AH) activity by ezetimibe monotherapy [46]. 

## 5. The Effect of PCSK9i on Platelet Function

PCSK9 is a protease involved in cholesterol metabolism in the liver [47]. Hepatocytes secrete PCSK9, which binds the LDLR receptor on the liver surface and causes its degradation. This leads to an increase in LDL-C cholesterol. PCSK9 inhibitors are monoclonal antibodies that reduce the expression of this protease, leading to a reduction in LDL-C cholesterol by about 60% and a reduction in major cardiovascular adverse events [48,49,50,51]. However, PCSK9 is produced and secreted by other cells: intestinal, pancreatic, adipocytes, kidney, and brain cells. Several recent studies have evaluated the involvement of PCSK9 in cardiovascular disease, independent of increased LDL-C [52]. High serum levels of PCSK9 were indicative of atherosclerotic disease. The clinical implication of this observation was an increased risk of ischemic events, even in subjects with low LDL-C levels. Interesting data demonstrate a correlation between increased serum PCSK9 and platelet reactivity. The PCSK9-REACT study described the association between PCSK9 and platelet activity in ACS patients treated with DAPT. High PCSK9 levels were associated with increased platelet reactivity (*p* = 0.004) and reduction in the effects of antiplatelet drugs. Furthermore, the results indicated a strong association between MACE and high PCSK9 levels [53]. The main limitation of the in vivo studies was the coadministration of statins and antiplatelets. Wang et al. evaluated the association between PCSK9 and platelet reactivity in vitro to exclude influences on the results of other drugs (statins and antiplatelets). Subsequently, the in vivo study in healthy subjects without statins and antiplatelets showed that subjects with high PCSK9 levels had greater platelet activation [54]. Franchi et al. evaluated the effects of evolocumab on platelet reactivity. Low LDL-C cholesterol levels were achieved at 30 days by all patients treated with evolocumab. While evolocumab was associated with a statistically significant reduction in platelet reactivity in the first 14 days, no difference in PRU was demonstrated compared to placebo at 30 days (*p* = 0.161) [55].

The Odyssey and Fourier studies demonstrated the direct effect of PCSK9i (alirocumab and evolocumab, respectively): the reduction in circulating LDL levels by 60%, independent of other hypolipidemic therapies taken by the patient. Moreover, both studies demonstrated a reduction in the primary endpoint in patients with high cardiovascular risk. Alirocumab and evolocumab demonstrated a reduction in cardiovascular events of 15% and 20%, respectively [53,54]. Gargiulo et al. collected real-life data in the AT-TARGET-IT registry, observing in 798 enrolled patients a median LDL-C reduction of 64.9% with high adherence to therapy (95.2%) [56]. A large meta-analysis by Guedeney et al. described a reduction in the risk of myocardial infarction, stroke, and myocardial revascularization at a mean follow-up time of 2.3 years [50]. 

Several studies have demonstrated an indirect effect on platelets through the stabilization of atherosclerotic plaque. The Huygens study demonstrated the protective effect of the hypolipidemic strategy on coronary plaques in patients with ACS. Plaques were characterized by optical coherence tomography (OCT) at baseline and 50 weeks after diagnosis of NSTEMI. Plaque stabilization was observed in 161 patients taking evolocumab, with an increase in FCT (+44.3% for placebo vs. +81.8% in the study drug group (*p* = 0.04) [57]. 

In addition, the PACMAN-AMI study demonstrated plaque regression, lipid core reduction, and increased FCT at 52 weeks after alirocumab administration. The evaluation was performed on plaques of noninfarct-related arteries by IVUS, OCT, and NIRS assessment. At 52 weeks, a reduction in the percentage volume of the atheroma (PAV) (−2.13% vs. −0.92%; *p* < 0.001) was described at the IVUS, a reduction in the maximum lipid load index (mean change within 4 mm was −79.42 with alirocumab and −37.60 with placebo, *p* = 0.006) and an increase in FCT (mean change 62.67 vs. 33.19 μm, *p* = 0.001) was described at the OCT [58]. 

In addition, a direct effect independent of LDL-C reduction has also been hypothesized. The protective effects can be explained by the effects of PCSK9i on cardiomyocytes, endothelial cells, smooth muscle cells, monocytes, and platelets (Table 2). 

Recent data have shown that PCSK9 production can occur directly at the level of endothelial cells and smooth muscle cells of vessels. At this level, PCSK9 binds CD36, LRP1, and LOX1 receptors, activating signal transduction pathways involved in the mechanism of inflammation, oxidative stress, lipid uptake, cell activation, and thrombogenesis [59]. 

The production of PCSK9 and Ox-LDL is increased in subjects with hypercholesterolemia. High circulating levels of PCSK9 activate platelet scavenger receptors CD36, which induce platelet activation via src and JNK kinases involved in the mechanism of thromboxane A2 production. PCSK9 increases ROS production via NOX2 activation on platelets. This increases Ox-LDL formation, which amplifies platelet activation via binding to the CD36 and LOX1 receptors. This triggers platelet aggregation and thrombogenesis through the expression of p-selectin, CD40L, and granule release. PCSK9is have been shown to reduce the activation of NOX2, CD36, and LOX1. In particular, they demonstrated a reduction in oxidative stress by reducing the NOX2 pathway [60]. Furthermore, platelet aggregation is also reduced, as a reduction in circulating platelet factors (thromboxane, PAF-4, CD62, CD40L, and p-selectin) is observed after 6 months of treatment [61].

## 6. The Effect of Inclisiran on Platelet Function

Inclisiran is a small interfering RNA (siRNA) directed against PCSK9. Unlike monoclonal antibodies, inclisiran penetrates hepatocytes and blocks the translation of PCSK9 mRNA. Consequently, more LDL receptors will be expressed on the surface of hepatocytes, removing LDL from the bloodstream. 

The ORION 9, 10, and 11 studies demonstrated a reduction in LDL-C of about 50% with only two administrations per year [62]. In addition, the ongoing ORION 4 and 5 studies are evaluating the impact of inclisiran on cardiovascular outcomes. The drug is administered subcutaneously, and the only reported adverse events are redness and itching at the injection site. The ORION-1 safety analysis showed that the drug did not cause immunogenicity towards platelets, immune system cells (lymphocytes and monocytes), and immune markers (IL-6 and TNF-α) [63]. 

PCSK9 is also involved in atherogenic mechanisms, independently of its action on LDL. This protein is involved in proinflammatory and prothrombotic effects. Furthermore, high levels of PCSK9 enhance platelet aggregation. Therefore, an indirect effect of inclisiran’s action on platelets is conceivable. Inclisiran, due to highly specific hepatic absorption, only acts at the hepatic level.

Therefore, inclisiran has no direct action for PCSK9 present in other tissues, such as intestinal, pancreatic, adipocytes, kidney, and brain cells [64]. 

## 7. The Effect of Triglycerides on Platelet Function

High triglyceride (TG) levels are a cardiovascular risk factor. Defects in genes controlling TG metabolism, such as APOC3, ANGPTL3, and ANGPTL4, have been linked to reduced TG and reduced CVD risk. In fact, in the Framingham Offspring Study, borderline elevated TG levels > 150 mg/dL were independently associated with a 10–20% increase in CVD [65]. The results of the retrospective TG-Real study assessed an increase in cardiovascular events and death in patients with elevated TG levels [66]. 

Polyunsaturated fatty acids (PUFA) form the basis of the therapeutic strategy for hypertriglyceridemia. In particular, PUFA-n3 (OMEGA 3) has demonstrated multiple effects, with anti-inflammatory and antithrombotic properties. An inverse relationship was observed between carotid intima-media thickness and PUFA-n3 use. The main components of PUFA-n3 are EPA (eicosapentaenoic acid) and DHA (docosahexaenoic acid). PUFA-n3 formulations are indicated, in combination with diet, to reduce serum triglyceride levels. Icosapent ethyl (IPE) is a drug recently approved by the FDA for cardiovascular prevention in subjects with serum triglycerides > 150 mg/dL. Its composition is 96% purified and stabilized ester of EPA. EPA and IPE have been shown to reduce CV events [67,68,69], in contrast to DHA which can cause increased LDL levels [70].

The REDUCE-IT study showed that 4 g/day of icosapent ethyl in patients with hypertriglyceridemia resulted in a 25% reduction in ischemic events and cardiovascular death. The indirect mechanism is due to the production of prostanoids (prostacyclin) with antiplatelet and vasodilator activity, anti-inflammatory activity, and the antioxidant capacity of PUFA-n3 through integration into mitochondrial and plasma membranes. Although no serious bleeding events have been demonstrated, increased bleeding episodes have been observed in subjects treated with EPA [71].

The potential interaction between EPA and antithrombotic drugs merits further investigation. The mechanism of the association between hypertriglyceridemia and platelet aggregation remains poorly understood. Nkambule et al. assessed platelet reactivity in 3416 patients divided into four groups according to blood triglyceride levels. In this study, platelet thrombogenicity was shown to be directly associated with triglyceride levels. Values considered normal (100–150 mg/dL) were associated with altered platelet aggregation. Specifically, in the 253 patients assigned to the high TG group (>200 mg/dL), platelet aggregation was significantly reduced, with increased thrombotic risk (*p* < 0.0001) [72]. The REDUCE-IT study showed that 4 g per day of icosapent ethyl in patients with hypertriglyceridemia resulted in a 25% reduction in ischemic events and cardiovascular death [73]. Further studies, however, are needed to clarify the role of icosapent ethyl on platelet activity.

## 8. Conclusions

Primary and secondary prevention of cardiovascular disease cannot be achieved without the use of drugs for dyslipidemia. The main mechanisms explaining the protective action of these drugs are related to mainly direct and perhaps indirect effects. The direct effects are linked to a reduction in circulating LDL cholesterol and triglycerides. Several studies have shown that the reduction in blood LDL-C is associated with a lower incidence of cardiovascular adverse events. Effects independent of blood LDL-C reduction are explained by the interaction of hypolipidemic drugs with the main mediators of the atherosclerotic process: platelets, endothelium, monocytes, and smooth muscle cells. Statins have also demonstrated a dose-dependent action on intimal hyperplasia, ventricular hypertrophy, and platelet activation. In particular, by interacting with specific platelet receptors, they reduce adhesion, aggregation, degranulation, inflammation, vasoconstriction, and oxidative stress. All these mechanisms explain the antithrombotic effect associated with statin therapy, which is amplified by the addition of ezetimibe. New drugs acting on the PCSK9 protein inhibition have demonstrated a potent effect on LDL levels with a parallel reduction in cardiovascular risk. In particular, evolocumab and alirocumab stabilize atherosclerotic plaque by reducing the lipid core and thickening the fibrous cap. They also interact with platelets, resulting in reduced platelet reactivity when added in fast track after ACS. However, this effect remains controversial as the initial reduction in platelet reactivity does not persist for longer observation times. In the REDUCE-IT trial, serious adverse events related to bleeding occurred in more patients in the icosapent ethyl group than in the placebo group (2.7% vs. 2.1% *p* = 0.06), suggesting a potential antiplatelet or anticoagulant effect [72]. Bempedoic acid acts by blocking adenosine triphosphate citrate lyase (ACLY), an enzyme of cholesterol synthesis. The results of the CLEAR study demonstrate an additional 20% reduction in LDL-C cholesterol. It also described a 13% reduction in cardiovascular events in primary and secondary prevention [74]. However, a clear effect of bempedoic acid on platelet has not been demonstrated [75].

In conclusion, the main effect of antidyslipidemic agents is mainly related to the reduction in LDL levels, which are causally related to atherosclerosis, vulnerable plaque, and triglycerides. Some studies suggested a potential role of these drugs also on platelet function. However, there is not yet clear evidence that the direct antithrombotic effects of these drugs may have a clinically beneficial impact. Further studies should be performed to address this grey zone.

## Figures and Tables

**Figure 1 ijms-24-11739-f001:**
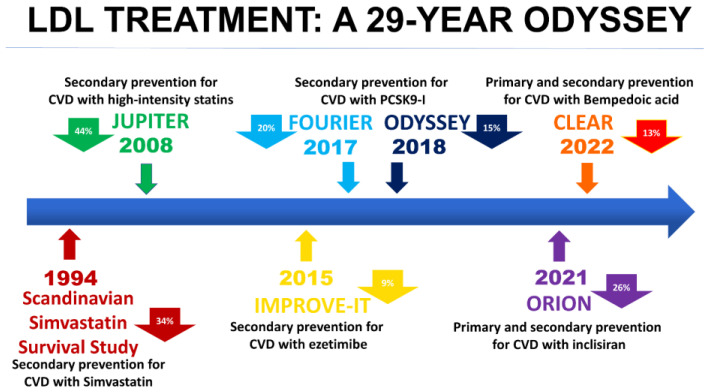
The evolution of hypercholesterolemic therapy. On the timeline, the figure shows the randomized clinical trial on hypercholesterolemia treatment. The value in the arrows indicates the reduction in the risk of cardiovascular events associated with the different treatments used. *CVD, cardiovascular disease; PCSK9i, proprotein convertase subtilisin/kexin type 9 inhibitors*.

**Figure 2 ijms-24-11739-f002:**
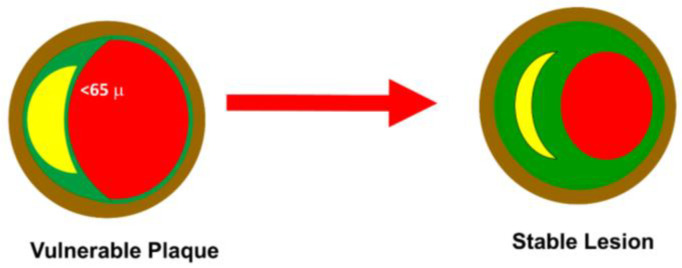
Plaque stability. A vulnerable plaque is defined as having an FCT of ≤65 μm with a pronounced lipid core. The anticholesterolemic drugs work through a reduction in lipid core and an increase in FCT. The severity of the lesion usually does not impact the transition from stable to unstable plaque.

**Figure 3 ijms-24-11739-f003:**
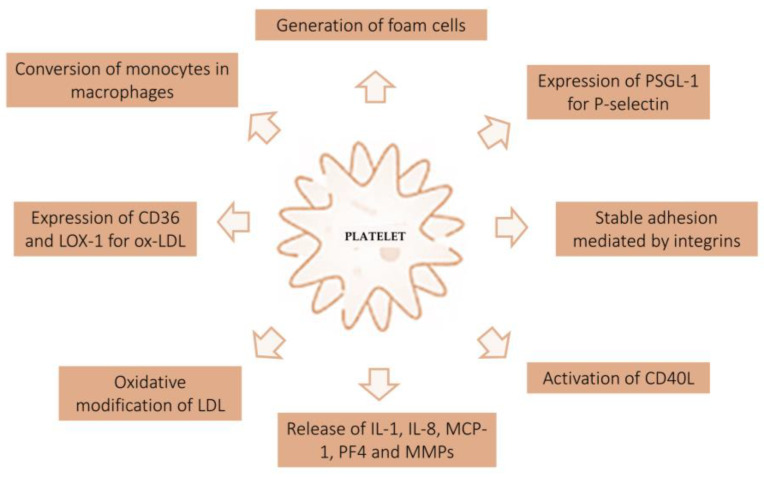
Interaction between atherosclerotic plaque and platelet activity. The figure shows the mechanism of platelet activation at the endothelial level. Based on in vitro and in vivo studies, platelets play a key role in the genesis of atherosclerotic plaque. *PSGL-1, P-selectin glycoprotein ligand-1; PF4, platelet factor 4; MCP-1, endothelial monocyte chemotactic protein-1; MMPs, metalloproteinases; LOX-1, lectin-like receptor-1*.

**Figure 4 ijms-24-11739-f004:**
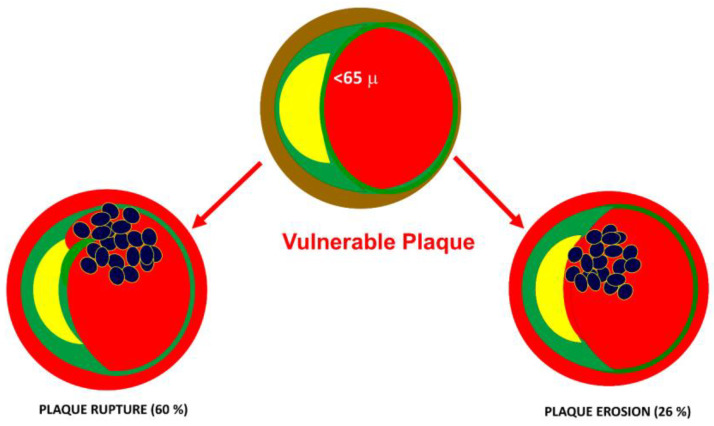
Underlying causes of acute coronary syndromes. The most frequent cause of acute coronary syndrome (ACS) is plaque rupture (60%). The second mechanism of ACS is the erosion of fibrous caps. In both cases, the exposure of the lipid core to the blood is the initial mechanism of platelet aggregation and intracoronary thrombus formation.

**Table 1 ijms-24-11739-t001:** Statin interaction in platelet function. The table summarizes the main interactions between platelet activity and the mechanism of action of statins. Inhibitory pathways are indicated in the red box. Activating mechanisms are indicated in the green box.

**Inhibition Pathway**	
P-selectin	Reduction in platelet stable adhesion
PAR-1	Reduction in platelet aggregation, release of proinflammatory particles, and production of adhesion molecules
TF	Reduction in extrinsic coagulation pathway and cMP release
CD40L	Reduction in serum inhibits platelet aggregation
NOX2/NADPH	Antioxidant effect
**Activation Pathway**	
PPARα/PPARƴ	Inhibition of platelet degranulation and aggregation by the suppression of PKCα pathway
	Increase in cAMP and decrease in Ca^2+^ level
eNOS	NO has vasoprotective, vasodilating effects, and reduced platelet aggregation
PLA2	Reduction in TxA2

PAR-1, protease-activated receptor-1; TF, tissue factor; cMP, circulating microparticles; NOX2, NADPH, PPARα/PPARƴ, peroxisome proliferator-activated receptor; PKCα, protein kinase Cα; cAMP, cyclic adenosine 3′,5′-monophosphate; eNOS, endothelial nitric oxide synthase; PLA2, phospholipase A2; TxA2, thromboxane A2.

**Table 2 ijms-24-11739-t002:** Pleiotropic effect of PCSK9 inhibitors. The table shows the effects of PCSK9i on cells involved in the pathogenetic mechanism of atherosclerosis. PCSK9-Is demonstrated a reduction in MACE in the randomized clinical trial.

Cardiomyocyte	EC	SMC	Macrophage	Platelet
Reduced autophagy	Reduced activation	Reduced proliferation and migration	Reduced accumulation in atherosclerotic plaque	Reduced activation
Reduce apoptosis	Reduced oxidative stress	Reduced oxidative stress	Reduced proinflammatory activation	Reduced aggregation
Improve function		Promotion of contractile phenotype	Increase cholesterol efflux	

PCSK9, proprotein convertase subtilisin/kexin type 9; EC, endothelial cell; SMC, smooth muscle cell.

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
