# Peer review of "The Effects of Statins, Ezetimibe, PCSK9-Inhibitors, Inclisiran, and Icosapent Ethyl on Platelet Function"

_ijms, 2023, doi:10.3390/ijms241411739_

Round 1
Reviewer 1 Report
The authors have presented a well written manuscript describing the Effects of Statins, Ezetimibe, PCSK9-inhibitors, Inclisiran 2 and Icosapent Ethyl on Platelet Function. I recommend the publication of this article in its present form and congratulate authors for their effort.
Reviewer 2 Report
The review is a good addition to the summary of potential antiplatelet effects of antidyslipidiemic agents.
The comments are below:
In the abstract, introduce what PCSK-9 is, e.g., in brackets, inhibitor of XX, also for icosapent.
In the introduction, after paragraph 54-59, a short paragraph explaining the mechanism of the LDL-C-lowering drugs mentioned in the article would be useful, including statins, ezetimibe, PCSK9-I, inclisiran, empedoic acid, etc.
Line 90-maybe "intact endothelium" instead of "intact endothelium"
Line 96-97: It is unclear what platelets or endothelium release (CD40, CD40L, IL8, MMP). Platelet activation increases the release of CD40L on the platelet surface, which sheds and then releases soluble CD40L. Activated platelets also express P-selectin.
Similar to lines 102-103, please reword or correct: Activated platelets bound to the endothelium release alpha-granules and secrete PF4 among other proteins.
Lines 108-110: please add which cells express class B and third class scavenger receptors
Tables 1 and 2 - please insert references.
Line 196 - "Statins are involved in the inhibition of P-selectin". Were the authors thinking of inhibition of exposure after cell activation? Please specify (it is not an enzyme).
Line 209, the statement "A key mechanism in plt aggregations..." is not correct. The key mechanism of platelet aggregation is via the binding of alphaIIbbeta3 integrin to fibrinogen. Were the authors thinking of platelet-endothelial binding under inflammatory conditions? Please correct.
The cAMP/PKA and cGMP/ NO pathways are the major platelet inhibitory pathways, and there are numerous downstream targets responsible for the inhibitory effect on platelets, please rephrase.
Please specify exactly in which cells statins cause an increase in eNOS. NOX2 is not a receptor, but an enzyme (subunit) of a complex that produces ROS. Please check and correct.
Line 253, "In vitro studies have shown a direct effect of ezetimibe on endothelium and platelets." - please cite the appropriate reference.
Line 292, please introduce here what evolocumab does.
Minor comments
Full name with the first mention of LDL-C, PCSK9-I, PCI, BMS - full names, and others in text, PGH2, PLA2, PAF-AH
Spelling: Plugs or Plaques? (Fig. 4 and text)
Minor editing of English language required, in rewording of some parts of text, as suggested above.
Reviewer 3 Report
Di Costanzo et al. reviewed in vitro and in vivo studies about the interplay of hypercholesterolemia, hypertriglyceridemia, platlet function and pharmacological treatments.
The review is very clear and well-writte, figures as self-esplicative.
I have just some minor revisions before recomending the publication of the manuscript:
Abstract: I would suggest to summarize the text included between lines 19-27 reporting just the main findings without including too specific details, that are already reported into the main text
Line 220, I think there is a typo, correct Ca into Ca2+
Reviewer 4 Report
Authors
Interaction between the dyslipidemia, platelet function, and related drug treatments is a relevant concept to study of cardiovascular disease. Increasing evidence support the view that statins, ezetimibe, PCSK9-inhibitors, inclisiran and icosapent ethyl, lipid-lowering therapy, also act as antithrombotic. Since the review refers to lipid-lowering therapy on platelet function, this later concept should better focus the rationale of the review in the main section of the manuscript. In addition, several recent reviews have partially covered the topic of interest (doi: 10.1007/s00228-021-03250-6; doi: 10.1016/j.ijcard.2022.11.026; doi: 10.1007/s12010-018-2776-5; doi: 10.1161/CIRCULATIONAHA.120.046290; doi: 10.3390/biomedicines9081073; doi: 10.1016/j.jacc.2020.12.058; doi: 10.1016/j.ahj.2021.01.018). To this respect, to increase novelty authors should focus the manuscript in discussing the most recent findings on key concepts in the field of antithrombotic regulation and lipid-lowering therapy and referring the readers to recent comprehensive reviews covering a more general overview of the topic.
Minor comments
The authors should include figures in the issue 2 to 7.
Figures 2 and 4 should show histological images, highlighting the tissue characteristics according to the atherosclerotic lesion.
